# Dynamic Analysis of Higher-Order Coordination in Neuronal Assemblies via De-Sparsified Orthogonal Matching Pursuit

**Shoutik Mukherjee and Behtash Babadi**
Department of Electrical and Computer Engineering
Institute for Systems Research
University of Maryland, College Park
College Park, MD 20742
{smukher2, behtash}@umd.edu

## Abstract

Coordinated ensemble spiking activity is widely observable in neural recordings and central in the study of population codes, with hypothesized roles including robust stimulus representation, interareal communication of neural information, and learning and memory formation. Model-free measures of synchrony characterize the coherence of pairwise activity, but not higher-order interactions; this limitation is transcended by statistical models of ensemble spiking activity. However, existing model-based analyses often impose assumptions about the relevance of higher-order interactions and require multiple repeated trials in order to characterize dynamics in the correlational structure of ensemble activity. To address these shortcomings, we propose an adaptive greedy filtering algorithm based on a discretized mark point-process model of ensemble spiking and a corresponding precise statistical inference framework to identify significant coordinated higher-order spiking activity. In the course of developing the statistical inference procedures, we also show that confidence intervals can be constructed for greedily estimated parameters. We demonstrate the utility of our proposed methods on simulated neuronal assemblies. Applied to multi-electrode recordings of human cortical ensembles, our proposed methods provide new insights into the dynamics underlying localized population activity during transitions between brain states.

## 1   Introduction

Coordinated ensemble spiking has been observed in a variety of brain areas, prompting a range of hypotheses about its role in cognitive function. Studies have documented synchronous spiking at all levels of the mammalian visual pathway [1, 2, 3]. Coordinated neural activity has additionally been hypothesized to influence interareal communication and the flow of neural information [4, 5, 6, 7, 8], and been postulated to be mediated by oscillations in local field potentials [9, 10, 11]. The study of synchrony is also closely tied to memory [12, 13, 14].

The prevalence of coordinated spiking and its functional implications for a range of neural processes have motivated both model-free and model-based approaches to its characterization. An intuitive model-free metric is the pairwise correlations of spike trains smoothed by a Gaussian (or exponential) kernel [15, 16]; several pairwise distance metrics have been proposed [17] as alternatives. Though the coherence of pairwise activity can be described, such measures do not capture higher-order coordination, and are limited in the ability to model dynamics in or determine the significance of pairwise coherence without repeated trials.

35th Conference on Neural Information Processing Systems (NeurIPS 2021).

Statistical models of neuronal ensemble activity transcend the limitation of model-free metrics to pairwise comparisons. Two widely-used approaches are the maximum entropy models and point process generalized linear models (GLM) [18, 19]. Maximum entropy models describe the state of the neural population only in terms of its instantaneous correlational structure [20, 21]. Models are estimated to match observed firing rates and all pairwise (and potentially higher-order) correlations simultaneously. Alternatively, point process GLMs for ensemble spiking [22, 23] characterize the influence of past population activity, or other relevant covariates. Though useful in estimating functional connectivity [24, 25], each neuron must be assumed conditionally independent due to regularity conditions that prohibit simultaneous spiking events [26, 27, 28]. This can be circumvented by using an equivalent marked point processes (MkPP) representation that explicitly models each disjoint simultaneous spiking event, as derived in [29] and expounded upon in [27]. MkPP representations of ensemble activity have also been utilized to analyze neuronal population coding in unsorted spiking data [30, 31]. A related approach models disjoint simultaneous spiking events as log-linear combinations of point process models, hence permitting an intuitive representation of excess or suppressed synchrony [10, 28].

Though statistical models can capture higher-order neural coordination, existing approaches face key limitations. Maximum entropy models can track dynamics in coordination using state-space filtering algorithms, but neglect the influence of past population activity on the ensemble state. Log-linear point process models address this shortcoming, but still share two shortcomings with maximum entropy models. First, assumptions on the relevance of higher-order interactions are typically imposed for tractable model estimation. Second, multiple repeated trials are required to capture dynamics in correlational structure and to evaluate the statistical significance of coordinated spiking.

We address these limitations by proposing an adaptive greedy filtering algorithm based on the discretized MkPP formulation in [27] to model dynamics in coordinated spiking within continuous recordings while capturing the influence of past ensemble activity. Furthermore, we build on recent theoretical results related to Adaptive Granger Causality analysis [24] to provide a precise statistical framework to detect significantly coordinated activity of arbitrary order. We demonstrate our proposed method's utility in tracking dynamics in coordinated spiking with statistical confidence on simulated ensemble spiking. Applying our method to continuous multi-electrode recordings of human cortical assemblies during anesthesia provides novel insights into coordinated spiking dynamics that underlie transitions between brain states.

## 2 Preliminaries

In this section, we review the discrete-time marked process representation and two corresponding likelihood models. First, we present a summary of key notation used throughout the subsequent sections in Table 1.

| Notation | Definition |
|---|---|
| $\boldsymbol{n}_t = \left[ n_t^{(1)}, \ldots, n_t^{(C)} \right]'$ | Ensemble spiking observation at time bin $t$ of $C$ neurons |
| $\lambda_t^{(c)} \Delta$ | Conditional Intensity Function (CIF) of $c^{\text{th}}$ neuron |
| $\boldsymbol{n}_t^* = \left[ {n^*}_t^{(1)}, \ldots, {n^*}_t^{(C^*)} \right]'$ | Marked observations at time bin $t$ of $C^* = 2^C - 1$ marks |
| ${\lambda^*}_t^{(m)} \Delta$ | CIF of $m^{\text{th}}$ mark |
| $n_t^{(g)}$ | Ground process, $\sum_{m=1}^{C^*} {n^*}_t^{(g)}$ |
| ${\lambda^*}_t^{(g)} \Delta$ | CIF of the ground process, $\sum_{m=1}^{C^*} {\lambda^*}_t^{(m)} \Delta$ |
| $\boldsymbol{\mu}_t = \left[ \mu_t^{(1)}, \ldots, \mu_t^{(C^*)} \right]'$ | Base rate parameters of mark events |
| $\boldsymbol{\omega}_t = \left[ {\boldsymbol{\omega}_t^{(1)}}', {\boldsymbol{\omega}_t^{(2)}}', \ldots, {\boldsymbol{\omega}_t^{(C^*)}}' \right]'$ | Model parameters of history-dependent model |
| $u_t^{(m)}$ | Log-odds of $m^{\text{th}}$ mark event vs. no spiking event |
| $u_{0,t}^{(m)}$ | Log-odds of $m^{\text{th}}$ mark event vs. no spiking event (restricted model) |
| $\gamma_t^{(m)} = u_t^{(m)} - u_{0,t}^{(m)}$ | Exogenous factor for $m^{\text{th}}$ mark |
| $\beta$ | Forgetting factor, $0 < \beta < 1$ |
| $W$ | Window length |

Table 1: Summary of key notation

## 2.1 Marked Process Representation of Ensemble Spiking

To characterize coordinated spiking, we utilize the discrete-time marked process (MkPP) representation of ensemble neuronal activity [27, 28]. For an ensemble of $C$ neurons, the $C$-variate spiking process, binned with small bin size $\Delta$, at time bin index $t$ is denoted by $\boldsymbol{n}_t := [n_t^{(1)}, n_t^{(2)}, \ldots, n_t^{(C)}]'$, where each component is the spiking process of one neuron. Conventional discrete point process models treat the components as conditionally independent Bernoulli observations. Given our interest in simultaneous spikes, we instead treat $\boldsymbol{n}_t$ as multivariate Bernoulli observations. The spiking process $\boldsymbol{n}_t$ is mapped to a $C^*$-variate process $\boldsymbol{n}_t^* := [n_t^{*(1)}, n_t^{*(2)}, \ldots, n_t^{*(C^*)}]'$, which are the binned observations of a marked point process whose marks count the number of exactly one of $C^* := 2^C - 1$ disjoint non-zero spiking events; we refer to $\boldsymbol{n}_t^*$ as the *marked* Bernoulli process, distinguishing it from the multivariate Bernoulli process $\boldsymbol{n}_t$. We define the mark space $\mathcal{K} := \{1, \ldots, C^*\}$ [26]. Fig. 1 shows an example of mapping the activity of $C = 3$ neurons to $C^* = 7$ marked processes. At each time $t_j$ such that $\boldsymbol{n}_{t_j} \neq \boldsymbol{0}$, the sole non-zero element of $\boldsymbol{n}_{t_j}^*$ indicates the mark. We also define the binned ground process $n_t^{(g)}$ that takes value 1 at each such $t_j$ and is zero otherwise [26]; the ground process indicates the occurrence of any spiking event and is represented by $n_t^{(g)} := \sum_{m=1}^{C^*} n_t^{*(m)}$.

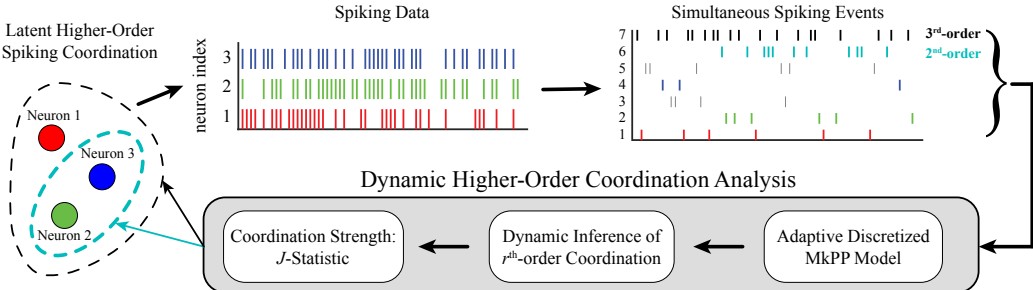

Figure 1: Ensemble spiking is mapped to a disjoint representation of simultaneous spiking events. The proposed method is used to infer the strength of higher-order coordination amongst $C$ neurons in a dynamic fashion.

The marked process representation is not unique, but can be defined in a convenient fashion: treating the components of $\boldsymbol{n}_t$ as the bits of a $C$-bit binary number, the mark indexed by the decimal equivalent of a particular realization of $\boldsymbol{n}_t$ will corresponded to that realization. By the disjointness of the marked representation, the spiking process of the $c^{\text{th}}$ neuron can be recovered as the sum of all marked process whose index, in binary, takes value 1 at the $c^{\text{th}}$ bit. For instance, in Fig. 1, the spiking data of neuron 3 (in blue) is the sum of simultaneous spiking event processes 4–7.

Our main contribution in this work is the dynamic and statistically precise inference of latent coordinated spiking of $C$ neurons using their simultaneous spiking representation (Fig. 1, bottom panel). To this end, we next describe two useful likelihood models for simultaneous spiking.

## 2.2 Two Likelihood Models of Simultaneous Spiking

In the discrete formulation, the conditional intensity functions (CIFs) of $\boldsymbol{n}_t$ and $\boldsymbol{n}_t^*$ are approximated by the probabilities of observing an event at time bin $t$ given the ensemble's spiking history. That is,

$$\lambda_t^{(c)}\Delta = \mathbb{P}[n_t^{(c)} = 1|\mathcal{H}_t], \qquad \lambda_t^{*(m)}\Delta = \mathbb{P}[n_t^{*(m)} = 1|\mathcal{H}_t], \tag{1}$$

for $c = 1, \ldots, C$ and $m = 1, \ldots, C^*$. We can relate $\lambda_t^{(c)}\Delta$ to $\lambda_t^{*(m)}\Delta$ in the same manner as $n_t^{(c)}$ to $n_t^{*(m)}$, and obtain the CIF of the ground process $\lambda_t^{*(g)}\Delta = \sum_{m=1}^{C^*} \lambda_t^{*(m)}\Delta$.

The marked process permits a generative description of simultaneous spiking events: ensemble spiking events are characterized by the ground process, occurring with probability $\lambda_t^{*(g)}\Delta$; the event is then assigned to the $m^{\text{th}}$ mark (i.e. the $m^{\text{th}}$ simultaneous spiking outcome) with conditional probability $\frac{\lambda_t^{*(m)}\Delta}{\lambda_t^{*(g)}\Delta}$. Thus, at time $t$ the likelihood of ensemble event $\boldsymbol{n}_t^*$ is given by:

$$p(\boldsymbol{n}_t^*) = \prod_{m=1}^{C^*} \left(\frac{\lambda_t^{*(m)}\Delta}{\lambda_t^{*(g)}\Delta}\right)^{n_t^{*(m)}} \left(\lambda_t^{*(g)}\Delta\right)^{n_t^{(g)}} \left(1 - \lambda_t^{*(g)}\Delta\right)^{1-n_t^{(g)}}. \tag{2}$$

The likelihood in (2) is used to form a multinomial generalized linear model (mGLM) with multinomial logistic link function of which we consider two versions. The first makes the simplifying assumption that there is no history dependence; the resulting model depends only on contemporaneous spiking, permitting compact parameterization by $\boldsymbol{\mu}_t = [\mu_t^{(1)}, \mu_t^{(2)}, \ldots, \mu_t^{(C^*)}]'$. Defining the baseline firing parameter for the $m^{\text{th}}$ mark to be

$$\mu_t^{(m)} := \log \left( \frac{\lambda_t^{*(m)} \Delta}{1 - \lambda_t^{*(g)} \Delta} \right), \quad m = 1, 2, \cdots, C^*, \tag{3}$$

or equivalently $\lambda_t^{*(m)} \Delta = \frac{e^{\mu_t^{(m)}}}{1 + \sum_{j=1}^{C^*} e^{\mu_t^{(j)}}}$, the log-likelihood can be rewritten as a linear function of $\boldsymbol{n}_t^*$, resembling the maximum entropy model [20, 21]:

$$\log p(\boldsymbol{n}_t^*) = \boldsymbol{\mu}_t' \boldsymbol{n}_t^* - \psi(\boldsymbol{\mu}_t), \quad \text{where} \quad \psi(\boldsymbol{\mu}_t) := \log \left( 1 + \sum_{m=1}^{C^*} e^{\mu_t^{(m)}} \right). \tag{4}$$

The second, more general version utilizes the ensemble history as covariates in the mGLM. Letting the covariate vector $\boldsymbol{x}_t$ be the ensemble history up to some fixed lag at time $t$ (augmented by a constant element of 1), the model is parameterized by $\boldsymbol{\omega}_t := [\boldsymbol{\omega}_t^{(1)'}, \boldsymbol{\omega}_t^{(2)'}, \ldots, \boldsymbol{\omega}_t^{(C^*)'}]'$, where the parameters for the $m^{\text{th}}$ mark $\boldsymbol{\omega}_t^{(m)} := [\mu_t^{(m)}, \boldsymbol{\theta}_t^{(m)'}]'$ consists of an ensemble history-modulation vector $\boldsymbol{\theta}_t^{(m)}$ in addition to the baseline firing parameter. Thus, the log-likelihood in this case admits a similar form to (4), by simply replacing $\mu_t^{(m)}$ with $\boldsymbol{x}_t' \boldsymbol{\omega}_t^{(m)}$.

## 3 Adaptive Estimation of the History-Dependent mGLM

Unlike conventional mGLM models, here the parameters are allowed to change in time. To capture their dynamics, we take a similar approach to the dynamic history-independent model of [32] and extend it to the history-dependent mGLM.

We assume conditional independence across time bins and that the parameters $\boldsymbol{\omega}_t$ admit piece-wise constant dynamics and are constant over consecutive windows of length $W$. The ensemble history up to lag $p$ defines the covariates as $\boldsymbol{x}_t := [1, n_{t-1}^{(1)}, \ldots, n_{t-p}^{(1)}, \ldots, n_{t-1}^{(C)}, \ldots, n_{t-p}^{(C)}]'$. The set of history covariate vectors at the $i^{\text{th}}$ window are denoted by $\boldsymbol{X}_i = [\boldsymbol{x}_{1+i(W-1)}, \ldots, \boldsymbol{x}_{iW}]'$. Since the mapping from $\boldsymbol{n}_t^*$ to $\boldsymbol{n}_t$ is injective, the influence of past spiking activity can be equivalently captured by defining history covariates in terms of either; however, using $\boldsymbol{n}_t$ reduces the dimensionality of $\boldsymbol{\omega}_t$ and quantifies the influence of past spiking activity directly rather than through categorical variables. Let $\boldsymbol{n}_i^{*(m)} = [n_{1+W(i-1)}^{*(m)}, \ldots, n_{iW}^{*(m)}]'$ denote the sequence of outcomes of the $m^{\text{th}}$ mark in the $i^{\text{th}}$ window. The log-likelihood of the $i^{\text{th}}$ window is thus given by

$$\ell_i(\boldsymbol{\omega}_i) := \sum_{m=1}^{C^*} \boldsymbol{n}_i^{*(m)'} \boldsymbol{X}_i \boldsymbol{\omega}_i^{(m)} - \sum_{j=1+(i-1)W}^{iW} \log \left( 1 + \sum_{m=1}^{C^*} e^{\boldsymbol{x}_j' \boldsymbol{\omega}_i^{(m)}} \right). \tag{5}$$

Motivated by the RLS objective function [33], a forgetting factor mechanism is utilized to combine the log-likelihoods up to the $k^{\text{th}}$ window, capturing the dynamics in each mark's rates. For a forgetting factor $0 \leq \beta < 1$, the adaptively-weighted log-likelihood at window $k$ is thus defined as:

$$\ell_k^{\beta}(\boldsymbol{\omega}_k) := (1 - \beta) \sum_{i=1}^{k} \beta^{k-i} \ell_i(\boldsymbol{\omega}_k). \tag{6}$$

Parameter estimation can be performed by solving a sequence of maximum likelihood problems:

$$\hat{\boldsymbol{\omega}}_k := \arg\max_{\boldsymbol{\omega}_k} \ \ell_k^{\beta}(\boldsymbol{\omega}_k), \quad k = 1, 2, \cdots, K. \tag{7}$$

Two issues arise when considering large ensembles. First, the dimensionality of $\boldsymbol{\mu}_k$ grows exponentially with $C$; second, it is likely that some marks will not contain any events. To address this, we employ a thresholding rule similar to [34], considering only "reliable interactions", i.e. the subset of the mark space $\bar{\mathcal{K}} = \{ m \in \mathcal{K} : \sum_t n_t^{*(m)} > N_{thr} \}$ for some pre-defined constant $N_{thr} > 0$, and treating the rates of the remaining marked processes as negligible due to their infrequency.

To efficiently solve the sequence of problems in (7) in an online fashion, we develop an adaptive greedy approach based on a generalized Orthogonal Matching Pursuit (OMP) [35] [36]. The OMP iteratively identifies the parameter support set (the non-zero components) of fixed size (a hyperparameter for which we cross-validate) to encourage sparsity, thus capturing the inherent sparsity of network interactions based on past ensemble activity [37, 38, 39]. Moreover, the greedy estimation over a sparse subset of parameters mitigates the intractability of the estimation problem for large ensembles where regularization-based constraints still require optimization over all parameters. The proposed adaptive OMP (AdOMP) algorithm, so named because the support set is permitted to change between windows, is detailed in Algorithm 1 in Appendix A.1. The key element of AdOMP is efficient evaluation of the gradient $\nabla_{\boldsymbol{\omega}} \ell_k^{\beta}(\boldsymbol{\omega}_k)$ at the $l^{\text{th}}$ iterate $\hat{\boldsymbol{\omega}}_{(l),k}$, to determine the next addition to the parameter support set and to solve the new maximization problem via gradient descent. Hence, its recursive computation is crucial for the algorithm to operate in an online fashion. To this end, we utilize a recursive update rule to compute the gradient at the $k^{\text{th}}$ window, generalizing the adaptive filtering techniques employed in [40] for Bernoulli observations to a multivariate setting.

# 4   Statistical Inference of Higher-Order Coordination

Coordinated spiking can indicate relationships between components of a neuronal ensemble and, potentially, effects of unobserved processes. However, simultaneous spiking events can still occur by chance amongst independent neurons, necessitating a test of significance to distinguish between excessive (or suppressed) and chance simultaneous events. In this section, we detail such a framework: first, we quantify the two alternatives by constructing a nested hypothesis test; second, we generalize the de-sparsifying procedure for $\ell_1$-regularized maximum-likelihood estimators established by [41] to the AdOMP; and third, we use the latter to establish a precise statistical inference framework by proving the applicability of an *adaptive de-biased deviance test*, used for identifying significant Granger-causal influences [24], to our setting.

## 4.1   Hypothesis Test Formulation for $r^{\text{th}}$-Order Coordinated Spiking

We characterize the significance of all $r^{\text{th}}$-order simultaneous spiking events for the history-dependent mGLM. The corollary for the history-independent model is addressed in Appendix C.4. The significance of $r$-wise simultaneous spiking for $r \geq 2$ is tested by considering the two alternatives:

$$
\begin{aligned}
H_0 \quad &: \quad r^{\text{th}}\text{-order simultaneous spikes occur as frequently as they would between} \\
&\quad\ \text{independent units, given ensemble spiking history} \\
H_1 \quad &: \quad r^{\text{th}}\text{-order simultaneous spikes occur at a significantly different rate than they} \\
&\quad\ \text{would between independent units, given ensemble spiking history}
\end{aligned}
\tag{8}
$$

A similar formulation is used in [28] to determine whether one mark occurs at a significantly different rate than expected. The likelihood of the mark is modeled as the product of marginal likelihoods times an additional multiplicative factor. Noting that the additional factor takes value 1 if the neurons are truly independent, the null hypothesis is quantified accordingly. To account for all marks of order $r$, we instead estimate a *reduced* model that assumes $r^{\text{th}}$-order interactions are chance occurrences by constraining the base rate parameters for each $r^{\text{th}}$-order mark. For the $m^{\text{th}}$ mark, let $u_t^{(m)} := \boldsymbol{x}_t{}' \boldsymbol{\omega}_k^{(m)} = \mu_k^{(m)} + \bar{\boldsymbol{x}}_t{}' \boldsymbol{\theta}_k^{(m)}$. We decompose the base rate parameter as $\mu_k^{(m)} = \mu_{0,k}^{(m)} + \gamma_k^{(m)}$, where $\mu_{0,k}^{(m)}$ is rate under the null hypothesis and $\gamma_k^{(m)}$ is analogous to the additional multiplicative factor in [28] that captures potential exogenous effects after conditioning on ensemble spiking history.

We thus estimate the *reduced* model $\hat{\boldsymbol{\omega}}_k^{(R)} := \arg\max_{\boldsymbol{\omega}_k^{(R)}} \ \ell_k^{\beta}(\boldsymbol{\omega}_k^{(R)})$, where the base rate parameters of $r^{\text{th}}$-order events are constrained to those under the null hypothesis. That is, for each $m \in \mathcal{K}_r := \{m \in \mathcal{K} : \sum_{c=1}^C m_c = r\}$, where $m_c$ is the $c^{\text{th}}$ least significant bit of $m$ in binary, we fix $\mu_k^{(m)}$ to $\mu_{0,k}^{(m)}$ and optimize the remaining parameters. To explicitly obtain the constraints, first recall that $\boldsymbol{x}_t{}' \boldsymbol{\omega}_k^{(m)}$ is the log-odds of $n_t^{*(m)} = 1$ versus $n_t^{(g)} = 0$ given ensemble spiking history. Under the assumption that the neurons are independent, the probabilities of each event is given, respectively, by

$$
\mathbb{P}[n_t^{*(m)} = 1 | \mathcal{H}_t] = \prod_{c_a : m_{c_a} = 1} \left( \lambda_t^{(c_a)} \Delta \right) \prod_{c_b : m_{c_b} = 0} \left( 1 - \lambda_t^{(c_b)} \Delta \right), \ \text{ and } \ \mathbb{P}[n_t^{(g)} = 0 | \mathcal{H}_t] = \prod_{c=1}^C \left( 1 - \lambda_t^{(c)} \Delta \right). \tag{9}
$$

Taking the ratio evaluated at the full model estimate $\hat{\boldsymbol{\omega}}_k$, we obtain $u_{0,t}^{(m)} := \sum_{c:m_c=1} \log\left(\frac{\hat{\lambda}_t^{(c)}\Delta}{1-\hat{\lambda}_t^{(c)}\Delta}\right)$. Assuming the difference between $u_t^{(m)}$ and $u_{0,t}^{(m)}$ is due only to exogenous factors, we estimate the corresponding term at the $k^{\text{th}}$ window as $\hat{\gamma}_k^{(m)} = \frac{1}{W}\sum_{t=(k-1)W+1}^{kW}\left(u_t^{(m)} - u_{0,t}^{(m)}\right)$. Thus, for the reduced model, we fix $\mu_k^{(m)}$ at $\hat{\mu}_k^{(m)} - \hat{\gamma}_k^{(m)}$ for $m \in \mathcal{K}_r$. The hypotheses are then quantitatively stated as:

$$H_0 : \boldsymbol{\omega}_k = \hat{\boldsymbol{\omega}}_k^{(R)}, \qquad H_1 : \boldsymbol{\omega}_k \neq \hat{\boldsymbol{\omega}}_k^{(R)}. \tag{10}$$

To control the possible abrupt variations of $\hat{\gamma}_k^{(m)}$ across windows, we apply Kalman forward/backward smoothing to the exogenous factor and use the smoothed values, $\tilde{\gamma}_k^{(m)}$, in lieu of $\hat{\gamma}_k^{(m)}$. The procedure is summarized in Algorithm 2 in Appendix A.3.

## 4.2 De-Sparsifying AdOMP Estimates

To test the hypotheses defined above, it is necessary to be able to construct confidence intervals for the parameter estimates. The procedure is well-established for unrestricted or unregularized linear regression models, but there is a paucity of work to this end for greedily-estimated high-dimensional sparse models. In the closely-related problem of $\ell_1$-regularized maximum-likelihood estimation, a set of elegant results [41, 42, 43] have established techniques to de-sparsify parameter estimates and construct confidence intervals. In particular, we extend the de-sparsification technique of [41], based on close inspection the Karush-Kuhn-Tucker conditions, in the greedy high-dimensional setting. In Appendix A.2, we derive the de-sparsified AdOMP parameters following $s^*$ iterations as

$$\hat{\boldsymbol{w}}_k := \hat{\boldsymbol{\omega}}_{(s^*),k} - \left(\nabla^2 \ell_k^{\beta}(\hat{\boldsymbol{\omega}}_{(s^*),k})\right)^{-1}\left(\nabla \ell_k^{\beta}(\hat{\boldsymbol{\omega}}_{(s^*),k})\right). \tag{11}$$

Next, the asymptotic normality of the de-sparsified AdOMP estimates is established. While a related result is established in [41], the independence of each realization of the covariates and observations is assumed; additionally, several conditions involved are tailored for $\ell_1$-regularized maximum likelihood estimation. Hence, we adapt the treatment in [41] for AdOMP to establish the following result:

**Theorem 1.** *Consider the maximization of the total data log-likelihood $\ell_k^{\beta}(\boldsymbol{\omega}_k)$ at the $k^{\text{th}}$ window, where the true parameter $\boldsymbol{\omega}_k \in \mathbb{R}^d$ is $(s,\xi)$-compressible with $\xi < \frac{1}{2}$. Let $\boldsymbol{\omega}_k^0$ be the maximum likelihood estimate and $\hat{\boldsymbol{\omega}}_k$ be the AdOMP estimate after $\mathcal{O}(s\log(s))$ iterations. If conditions (C1)–(C6) are met, the de-sparsified AdOMP estimate $\hat{\boldsymbol{w}}_k$ satisfies*

$$\sqrt{\frac{1+\beta}{1-\beta}}\left(\hat{\boldsymbol{w}}_k - \boldsymbol{\omega}_k^0\right) = \boldsymbol{V}_k + o_{\mathbb{P}}(1)\cdot\boldsymbol{1},$$

*where, as $\beta \to 1$, $\boldsymbol{V}_k \xrightarrow{d} \mathcal{N}(\boldsymbol{0}, \mathcal{I}_k^{-1})$ with $\mathcal{I}_k^{-1} = -\Sigma_k^{-1}$ the inverse of the Fisher information matrix.*

For brevity, the technical conditions (C1)–(C6) are omitted here, but are presented in Appendix C.2 along with a detailed proof. Based on Theorem 1, confidence intervals for the AdOMP estimates can be constructed by adapting the recursive procedure of [40] to our setting.

## 4.3 Deviance Difference Test for $r^{\text{th}}$-Order Coordinated Spiking

Classical results on likelihood ratio tests between two nested hypotheses [44, 45] have established the use of the deviance difference as a common procedure. However, they are ill-suited in our setting due to the highly-dependent covariates and forgetting-factor mechanism in the data log-likelihood. These issues are addressed in a related context [24] for the inference of Granger-causal links by defining the *adaptive* de-biased deviance difference and characterizing its limiting distribution under presence and absence of Granger-causal links. We similarly utilize the adaptive de-biased deviance difference

$$D_{k,\beta}^{(r)}\left(\hat{\boldsymbol{\omega}}_k^{(F)}, \hat{\boldsymbol{\omega}}_k^{(R)}\right) := \left(\frac{1+\beta}{1-\beta}\right)\left[2\left(\ell_k^{\beta}(\hat{\boldsymbol{\omega}}_k^{(F)}) - \ell_k^{\beta}(\hat{\boldsymbol{\omega}}_k^{(R)})\right) - \left(\mathscr{B}_k^{(F)} - \mathscr{B}_k^{(R)}\right)\right] \tag{12}$$

as the test statistic, where $\mathscr{B}_k^{(F)}$ and $\mathscr{B}_k^{(R)}$ are the respective biases of the full and reduced models. As we show in Appendix A.1, the full and reduced log-likelihoods can also be computed in an online fashion, in a similar manner as the gradients.

The limiting distributions of the adaptive de-biased deviance difference for the greedily-estimated joint model under both the null and alternative hypotheses are characterized in a similar fashion to [24] by utilizing the asymptotic normality of the de-sparsified AdOMP estimates from Theorem 1:

**Theorem 2.** *Let $\hat{\boldsymbol{\omega}}_k^{(F)}$ and $\hat{\boldsymbol{\omega}}_k^{(R)}$ respectively be the full and reduced greedily-estimated mGLM parameters at window $k$, where $\hat{\boldsymbol{\omega}}_k^{(R)}$ assumes conditionally independent $r^{\text{th}}$-order simultaneous spiking. Then, as $\beta \to 1$,*

   *i) if $r^{\text{th}}$-order coordination matches independent $r^{\text{th}}$-order interactions given ensemble spiking history, then $D_{k,\beta}^{(r)}(\hat{\boldsymbol{\omega}}_k^{(F)}, \hat{\boldsymbol{\omega}}_k^{(R)}) \xrightarrow{d} \chi^2(M^{(r)})$, i.e. chi-square, and*

   *ii) if $r^{\text{th}}$-order coordination diverges from independent $r^{\text{th}}$-order interactions given ensemble spiking history, and assuming the base rate parameters of $r^{\text{th}}$-order interactions scale at least as $\mathcal{O}\left(\sqrt{\frac{1-\beta}{1+\beta}}\right)$, then $D_{k,\beta}^{(r)}(\hat{\boldsymbol{\omega}}_k^{(F)}, \hat{\boldsymbol{\omega}}_k^{(R)}) \xrightarrow{d} \chi^2(M^{(r)}, \nu_k^{(r)})$, i.e. non-central chi-square,*

*where $\nu_k^{(r)}$ is the non-centrality parameter at window $k$ and the degrees of freedom $M^{(r)} := |\mathcal{K}_r|$ is the difference in the cardinalities of the full and reduced support sets.*

A detailed proof is provided in Appendix C.3. In order to fully characterize the limiting distribution of $D_{k,\beta}^{(r)}$ under $H_1$, we must estimate the non-centrality parameter for each window. Assuming the parameter evolves smoothly in time, we use a state-space smoothing algorithm [24] to estimate it from the observed $D_{k,\beta}^{(r)}$ values. This not only allows us to identify significant coordination, but to also quantify the degree of significance using Youden's $J$-statistic

$$J_k^{(r)} := 1 - \alpha - F_{\chi^2(M^{(d)}, \hat{\nu}_k^{(r)})}\left(F_{\chi^2(M^{(d)})}^{-1}(1 - \alpha)\right) \tag{13}$$

for significance level $\alpha$, where $F(\cdot)$ denotes the CDF. Values of $J_k$ close to $1$ imply that the rejection of the null is a stronger indication of coordination than for smaller values of $J_k$. Thus, the $J$-statistic characterizes the test in terms of both type I and type II errors. By convention, we take $J_k = 0$ when $H_0$ is not rejected at the $k^{\text{th}}$ window. Under the alternative, it is possible to observe either significant excess or suppressed coordination; this can be reflected in the $J$-statistic by incorporating the net exogenous effect on $r^{\text{th}}$-order coordination and using a signed $J$-statistic $J_k^{(r)} \cdot \text{sgn}\left(\sum_{m \in \mathcal{K}_r} \hat{\gamma}_k^{(m)}\right)$. The full procedure for identifying significant $r^{\text{th}}$-order coordinated spiking is summarized by Algorithm 3 in Appendix C.

## 5 Applications

### 5.1 Simulated Ensemble Spiking Data

We validate our proposed methods in a simulated example, performing complementary history-independent and history-dependent analyses. Let the base rate parameter and exogenous effect for the history-independent model be denoted by $\mu_k$ and $\gamma_k$; and the same for the history-dependent model by $\mu_{k,\mathcal{H}}$ and $\gamma_{k,\mathcal{H}}$, with history-modulation parameter $\boldsymbol{\theta}_k$. Then, the reduced model constraints imply $\gamma_k = \gamma_{k,\mathcal{H}} + \bar{\boldsymbol{x}}_t' \boldsymbol{\theta}_k$. If the observed rate of higher-order events is equal to that of independent neurons, $\gamma_k = 0$; however, higher-order interactions may still be coordinated, i.e. $\gamma_{k,\mathcal{H}} = -\bar{\boldsymbol{x}}_t' \boldsymbol{\theta}_k \neq 0$. Conversely, the observed rate of higher-order events may differ from that of independent neurons, i.e., $\gamma_k \neq 0$. If $\gamma_{k,\mathcal{H}} = 0$, observed coordination can be attributed to the effects of ensemble history; otherwise, observed coordination was driven by an unobserved process. A MATLAB implementation of both algorithms is provided in supplementary material.

Ensemble spiking of five neurons was generated by a marked Bernoulli process as described in Eq. (2) with an average rate of $\sim 0.1$ spikes per bin. In the first and third simulated epochs, $4^{\text{th}}$-order spiking events were excited by amplifying the default history-modulation parameters. In the second epoch, the base rate parameter was increased to induce $3^{\text{rd}}$-order spiking events. These adjustments respectively reflected simultaneous spiking induced by ensemble history and by an unobserved process. Figure 2–A shows the simulated spiking activity, from which no obvious coordination is observable. The aggregate $r^{\text{th}}$-order marks are visualized in Fig. 2–B, with apparent increased rates of $3^{\text{rd}}$- and $4^{\text{th}}$-order spiking events.

For comparison, we also used three single-trial measures of coordinated spiking. The first is the average Pearson correlation between smoothed spiking responses. The second, is the spiking regularity, quantified by average coefficient of variation (ratio of the standard deviation to the mean inter-spike interval) [46]. A ratio close to $1$ indicates Poisson statistics; larger ratios indicate greater

variability due to self-exciting dynamics while smaller ratios indicate regularity in spiking (i.e. globally coordinated spiking). Both measures are computed over non-overlapping windows of 250 samples to track dynamics. The third measure is the average difference between $r^{\text{th}}$-order mark CIFs and probabilities of $r^{\text{th}}$-order independent interactions, generalizing the measure employed in [27] to higher-order simultaneous spiking. Other model-based analyses require multiple trial repetitions and are thus unsuited to our single-trial simulation setting.

Statistical analyses of $r^{\text{th}}$-order coordination for $r = 2, \ldots, 5$ using the history-independent model ($W = 10$; $\beta = 0.975$) reveals facilitated $3^{\text{rd}}$-order coordination during the second epoch, indicated by large positive values of the $J$-statistics (Fig. 2–C). Facilitated $4^{\text{th}}$-order coordination is detected during the first and third epochs. Ensemble spiking was also analyzed using the history-dependent model ($W = 10$; $\beta = 0.99$) (Fig. 2–D). Conditional facilitation of $3^{\text{rd}}$-order coordination was correctly detected during the second epoch and $4^{\text{th}}$-order coordination was correctly conditioned out. The history-dependent analysis also detected conditional suppression of $2^{\text{nd}}$-order coordination.

In contrast, the three control measures are unable to capture the underlying dynamics. Significant pairwise correlations (Fig. 2–E) are stably indicated throughout the simulation, insensitive to changes in coordinated spiking across epochs. Similarly, the spiking regularity (Fig. 2–F) indicates Poisson spiking statistics rather than coordinated activity. The $3^{\text{rd}}$- and $4^{\text{th}}$-order mark CIF differences (Fig. 2–

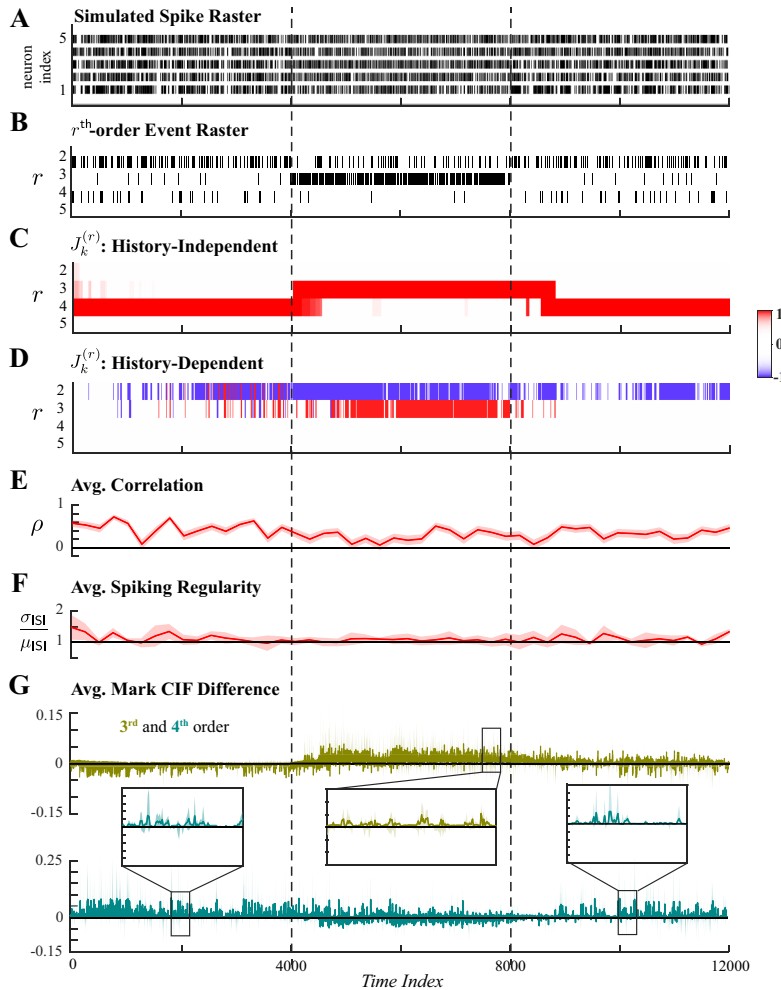

Figure 2: Analysis of ensemble spiking with non-overlapping epochs of $3^{\text{rd}}$- and $4^{\text{th}}$-order coordination. **A.** Simulated ensemble spiking of five neurons. **B.** Sum of the $r^{\text{th}}$-order simultaneous spiking events for $r = 2, 3, 4, 5$. Spiking coordination varies across 3 epochs, demarcated by vertical dashed lines. **C.** Significant $r^{\text{th}}$-order coordination neglecting ensemble history. **D.** Significant $r^{\text{th}}$-order coordination based on history-dependent ensemble spiking model. Statistical testing in **C–D** performed at level $\alpha = 0.001$. **E.** Average Pearson correlation with $95\%$ confidence interval. **F.** Average spiking regularity: coefficient of variation $\pm 2$ SEM. **G.** Average mark CIF differences of $3^{\text{rd}}$- (green) and $4^{\text{th}}$-order (teal) spiking interactions $\pm 2$ SEM.

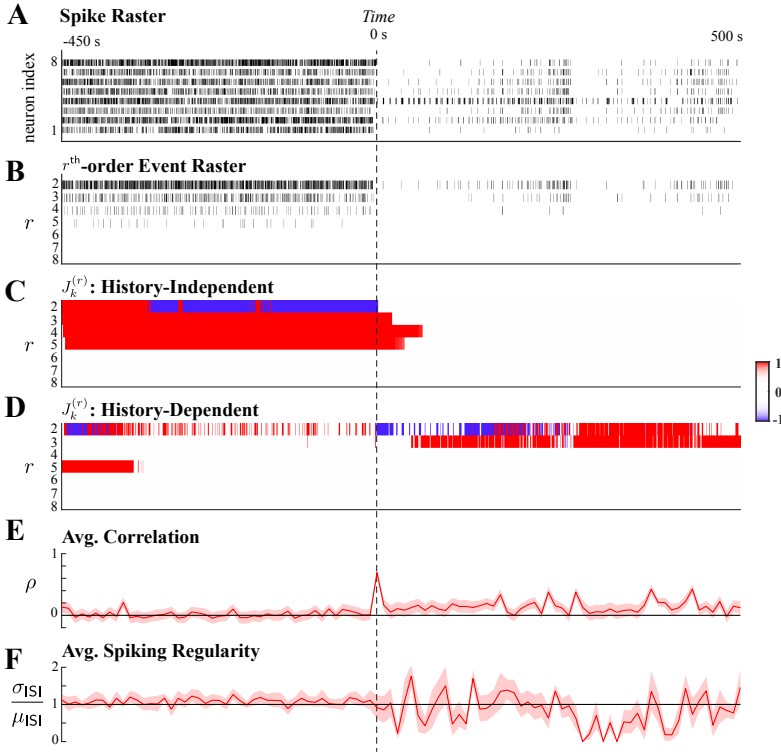

Figure 3: Coordinated spiking in human cortical neurons is exogenously induced during unconsciousness. **A.** Raster of 8 cortical neurons aligned to loss of consciousness (LOC) at 0 s. The anesthetic is administered twice: at 0 s and at ∼250 s. **B.** Sum of the $r^{\text{th}}$-order simultaneous spiking events for $r = 2, \ldots, 8$. **C.** Significant $r^{\text{th}}$-order coordination neglecting ensemble history. **D.** Significant $r^{\text{th}}$-order coordination based on history-dependent analysis. Statistical testing in **C–D** performed at level $\alpha = 0.001$. **E.** Average Pearson correlation with $95\%$ confidence interval. **F.** Average spiking regularity: coefficient of variation $\pm 2$ SEM.

G) weakly reflect the underlying dynamics, but closer inspection reveals the oscillatory nature of this sample-to-sample measure that diminishes its reliability (Fig. 2–G, insets). We further examine the effect of hyperparameters $W$ and $\beta$ in Appendix B.1, inspect the model goodness-of-fit in Appendix B.2, and provide a complementary simulation study with more complex dynamics in Appendix B.3.

## 5.2   Real Data Example: Anesthesia Data

We next present our analysis of human cortical neuronal assemblies during the transition into propofol-induced general anesthesia. The data were retrieved in a fully anonymized format with permission from the authors in [47], who obtained written consent from the participants in compliance with the institutional review board (please refer to [47] for details). We employed the proposed algorithms to analyze higher-order coordination and compared them against the average Pearson correlation and spiking regularity. The CIF-based measure is omitted given its highly oscillatory nature, rendering its interpretation uncertain. We analyzed spiking data from one subject, selecting the 8 neurons with the highest average firing rate. In contrast to the simulation in Fig. 2, the average firing rate was $\sim 0.05$ spikes per bin. In Fig. 3–A, their ensemble spiking activity is shown, aligned to the loss of consciousness (LOC) at 0 s. Ensemble activity recovered after ∼250 s, when propofol was re-administered. The decomposition of ensemble spiking into $r^{\text{th}}$-order events (Fig. 3–B) highlights lower rates of higher-order spiking events.

History-independent analysis ($W = 10$; $\beta = 0.99$) of $r^{\text{th}}$-order spiking revealed high rates of $3^{\text{rd}}$-, $4^{\text{th}}$-, and $5^{\text{th}}$-order events during consciousness (Fig. 3–C). Additionally, $2^{\text{nd}}$-order coordination was suppressed during consciousness. However, history-dependent analysis ($W = 10$; $\beta = 0.995$) did not identify conditional coordination during consciousness (Fig. 3–D). Together, these suggest that rates of higher-order simultaneous spiking events diverged from the rate of such interactions amongst independent neurons, but coordination during consciousness is attributable to ensemble history. The

high $J$-statistic values at the start of the recording were transients related to the initial convergence of the adaptive filters.

After LOC, the rates of simultaneous events matched those amongst independent neurons (Fig. 3–C). However, significant $2^{nd}$- and $3^{rd}$-order conditional coordination were detected during anesthesia. Third-order coordination was persistently and exogenously facilitated; $2^{nd}$-order coordination changed from a state of initial suppression to facilitation at $\sim 250$ s. Our results show local neuronal networks during both conscious and unconscious states exhibited coordinated spiking, but the underlying mechanisms differed between states, and the differences manifested rapidly in concurrence with LOC.

These dynamics are poorly reflected by the Pearson correlation and spiking regularity, both computed over windows of 250 samples (resp., Fig. 3–E and Fig. 3–F). Prior to LOC, pairwise correlations may have been high but spiking was not globally synchronized. Hence, spiking statistics seem to match Poisson spiking and neurons seem uncorrelated because of irregular low-order coordinated spiking. Slight increases in correlation during anesthesia weakly indicate coordinated spiking. The fluctuating spiking regularity leaves the nature of higher-order coordination indeterminate.

# 6   Concluding Remarks

The proposed modeling and statistical inference algorithms constitute a novel approach to studying coordinated neuronal spiking. In contrast to previous model-based approaches, the proposed method is tailored for the analysis of continuous recordings of neuronal data. We demonstrated that the framework can capture both time-varying spiking rates and the influence of spiking history, and thus can detect endogenously or exogenously induced coordinated spiking.

In developing this framework, we showed that confidence intervals can be constructed around greedily estimate parameters in similar fashion to sparsity-regularized parameter estimates. We found this to be a noteworthy gap in existing literature, as theoretical analyses of greedy algorithms focused instead on guarantees of model recovery. This result enabled us to develop a precise statistical inference framework in which the statistical strength of discovered synchronous spiking can be quantified.

Simulation studies demonstrated the efficacy of our framework in detecting suppressed or facilitated coordinated spiking activity. Moreover, in application to spontaneous ensemble spiking during the transition into propofol-induced anesthesia, our proposed method provided greater detail about the correlation structure of local neuronal networks in both the conscious and unconscious states. Additionally, our results reflected the abruptness of the transition between network states by characterizing dynamics in coordinated spiking. The ability to track transitions in higher-order network interactions through adaptive filtering techniques can be used to address current gaps in understanding the local mechanisms underlying the emergence of different brain states.

## Acknowledgements

This work is supported in part by the National Science Foundation awards no. 1552946 and 2032649 and the National Institutes of Health award no. 1U19NS107464.

The authors declare no competing financial interests.

We thank Emery N. Brown and Patrick L. Purdon for providing neural data (published in [47]) recorded from human cortical assemblies during anesthesia.

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
