# OpenReview forum: "Dynamic Analysis of Higher-Order Coordination in Neuronal Assemblies via De-Sparsified Orthogonal Matching Pursuit"
_NeurIPS.cc/2021/Conference — NeurIPS 2021 Poster_

### Official Review · Reviewer_nqVp · 2021-07-16

**Rating:** 7
**Confidence:** 3

**Summary:**

The current manuscript presented an algorithm based on marked point processes to distill higher-order interactions in neuron ensembles. The authors provided thorough theory for the algorithm and demonstrated the efficacy of the algorithm in detecting both time-varying spiking rates and coordinated spiking components in simulated and real neuron ensembles.

**Limitations And Societal Impact:**

The authors did not discuss the limitation of current work. I think one of the major limitations of the current model is lack of functional foundation or biology meaning of model components.

**Main Review:**

In general, the authors presented a novel approach for analyzing higher-order interactions in a neuron ensemble. The simulation results suggested that the algorithm successfully identify the endogenous components of higher-order coordinated spikes. In application of real neuron ensemble, the algorithm identified brain state modulated higher-order interactions. The paper was well written, and the experiments were carefully designed to demonstrated the power of the current algorithm. However, I would concern about the scope of application of this model. First, the anesthesia data has relatively simple structure, that single factor (anesthesia) generated large effects on neuron spiking rates. And the authors only tested the model on eight neurons with the highest spiking rate from the ensemble. It is unclear whether the model would converge for a data set with more complication structure and lower spiking rate. Second, the identified higher-order interactions and the history-dependence did not readily interpretable, as the modeling components that generated this knowledge did not have a biology meaning. In many real-world applications, users would not be able to tell whether identified higher-order interactions and history dependence were real or meaningful.

**Time Spent Reviewing:**

5

---

> ### Author Response · Authors · 2021-08-10
> **We will revise the manuscript to address the effects of spiking rates and latent dynamics, and to address the reviewer's questions about the biological implications of our statistical model.**
>
> - **(Regarding effects of spiking rates and complexity of latent dynamics):** We agree with the reviewer that these are important considerations for applying our proposed analyses. We note that the eight neurons analyzed during anesthesia had a low average firing rate of 0.0547 spikes/bin (equivalently, 1.0941 Hz). In contrast, our simulated neuronal ensemble had an average firing rate of 0.0972 spikes/bin. The text will be revised to emphasize that the effects of average spiking rate have been considered. We will add an additional supplementary simulation with more complex latent dynamics and a larger ensemble size for concreteness.
> - **(Regarding interpretability):** Thank you for identifying the interpretation of our analyses as a point for clarification. As the reviewer suggested, our generative model is not biological. It is closely related to the maximum entropy model (note Equation 4 near line 102), which is inspired by statistical physics, and makes minimal assumptions about the underlying interactions to capture time-varying coordinated activity in an ensemble of spiking neurons. The identified higher-order coordination, however, can provide biological hypotheses on the circuit mechanisms involved. For instance, the dynamics of the higher-order coordination observed in the history-dependent analysis of the anesthesia data suggests the involvement of an exogenous (i.e. non-cortical) process in spiking coordination. This process indeed is hypothesized to originate from thalamic inputs to the cortex under propofol-induced anesthesia. As Reviewer 1 has also suggested, we will revise the discussion of our analyses to make it more accessible to a broader readership, particularly clarifying its limitations as a statistical model (vs. a biological model) and discussing its biological implications in suggesting potential circuit mechanisms.

---

### Official Review · Reviewer_KgmN · 2021-07-16

**Rating:** 5
**Confidence:** 3

**Summary:**

The authors describe a time-varying multinomial GLM model which detects higher-order correlations in multi-dimensional spike trains. See main review for further summary.

**Limitations And Societal Impact:**

The limitations of the work and motivation behind modeling choices could be more clearly and concisely discussed. I foresee no potential negative societal impacts.

**Main Review:**

It took me a while to understand the contributions made in this paper, but I think they can be summarized as extending the multinomial GLM model of Ba et al. (2014, Frontiers in Comp Neuro) to have time-varying parameters. This extension leans heavily on methods described in Sheikhattar et al. (2015, IEEE Transactions on Signal Processing). The goal of this new model is to capture higher-order spike time correlations between simultaneously recorded neurons. The new method does not require repeated trials.

My biggest concern with this paper is the difficulty I had in understanding the motivations behind various modeling choices and implementation details. I feel like I only began to understand this paper after reading Ba et al. (2014) and Sheikhattar et al. (2015) in careful detail.

For example, a window length W and a forgetting factor \beta control how quickly the model parameters vary over time. If the forgetting factor is close to zero and the window lengths are relatively small, then the parameters can vary rapidly, potentially resulting in overfitting -- how is this avoided? If the forgetting factor is close to one, then the parameters vary only very slowly, potentially resulting in an underfit model. Even if the forgetting factor is well tuned, this model can capture a gradual drift in parameters, but potentially it will have trouble accounting for rare, step-like changes in network state, as happens during propofol induced loss of consciousness (investigated in Fig 3). The paper would be stronger if the dynamics of the time-varying parameters were more clearly discussed and illustrated in practice.

The assumption of sparsity on the GLM parameters is not clearly motivated or discussed. Why not use l2 regularization instead of greedy sparse estimates? The Orthogonal Matching Pursuit algorithm is not reviewed, and terms like "support set" (refering to the subset of nonzero parameters) are used without a clear definition. On lines 171-172 the authors say they apply a Kalman forward filter and backward smoother control "abrupt variations" but the details of this smoothing are not present in the main or supplemental text.

The application of the model to real data is discussed only in a short section (5.2). How were the \beta parameters selected in this analysis? How much sparsity was imposed on the parameter estimates, and how was this tuned?

Despite these misgivings, I think the model addresses an interesting topic and I am open to raising my score if the other reviewers were better able to understand the details of the model.


Other Comments
--------------

- On line 318 there is a typo, "estimate" should be "estimated"

- In figures 2 + 3, please label the blue-to-red colorbar

- On lines 167-169, are the "u" variables meant to be "\mu"?

**Time Spent Reviewing:**

4

---

> ### Author Response · Authors · 2021-08-10
> **We will revise the manuscript to more clearly motivate modeling choices and include additional simulations to address the reviewer's questions about their effects on model goodness of fit.**
>
> - **(Regarding model fit vs. hyperparameters $W$ and $\beta$ ):** We agree with the reviewer on the merits of clearly address hyperparameter selection and goodness of fit. We will include supplementary simulations that address the expected effects of varying window length and the forgetting factor. These will include quantitative measures of model goodness of fit. Invoking the multivariate extension of the point process time-rescaling theorem, the Kolmogorov-Smirnov (KS) test is used to compare rescaled empirical interspike intervals -- of each mark event -- to uniform quantiles. Additionally, the autocorrelation function (ACF) is used to test if rescaled interspike intervals are uncorrelated. Representative examples of the KS and ACF tests for multiple marks will be included in the revision.
> - **(Regarding motivations for greedy sparse model estimation):** We will revise the text so that our justification for imposing sparsity on mGLM parameters through the Orthogonal Matching Pursuit is clearer. In brief, the motivation was two-fold.
>     - First, network interactions based on past ensemble spiking history are hypothesized to be sparse, as supported by several empirical and theoretical studies that we will be sure to reference in our revision. Hence, a history-dependent model of coordinated ensemble spiking events should account for this sparsity.
>     - Second, optimizing over all model parameters (as would be the case for regularized optimization) becomes intractable as the number of neurons analyzed increases. Greedy approaches mitigate this issue by optimizing over a small subset of these parameters (i.e. the support set, which is defined as the subset of non-zero parameters). The Orthogonal Matching Pursuit identifies the support set iteratively rather than it being fixed beforehand. As such, if the true model turns out to be dense, the OMP procedure will select a large support set (via cross-validation) and adapts to the dense support of the parameters. As recommended by Reviewer 1, we will include a preliminary discussion of mathematical notation and nomenclature in which we will explicitly define the relevant terminology.
> - **(Regarding details of Kalman F/B smoothing):** Thank you for this recommendation; we will include the motivation for and details of the forward/backward smoothing algorithm in supplemental text.
> - **(Regarding hyperparameters for real-data analysis):** The parameter $\beta$ was selected based on the "effective" integration window length given by $W/(1 - \beta)$, as established by Sheikhattar et al. (2015). For the selected values of $W$ and $\beta$, this ratio corresponded to a 100 second window; this selection enabled us to capture the effects of slow oscillations ($\sim$ 0.1 Hz) over multiple periods while still permitting the tracking of transitions between brain states. We cross-validated for the size of the support set that maximized the static log-likelihood, thereby assuming the sparsity to be constant for simplicity. We will provide more details about the choice of $\beta$ and cross-validation procedure for choosing the sparsity of the mGLM parameters.
> - **(Regarding typos and figure colorbars):** Thank you for pointing out these errors. We will correct the typo and amend the figure.
> - **(Regarding notation on lines 167-169):** Thank you for pointing out this lack of clarity -- the variables are correctly referenced in the text, but we will revise the text so that these quantities are more clearly defined in a preliminary subsection on notation. The variables "$u$" are the modulations of the mark CIFs, and stand for the argument of the logistic function as defined on line 158. For the null hypothesis, "$u\_{0}$" is computed from the CIFs of the spiking processes rather than the inverse logistic function.

---

> > ### Comment · Reviewer_KgmN · 2021-08-18
> > **Quick question**
> >
> > Thank you for your reply. I wanted to ask a couple of quick questions regarding motivation for the model.
> >
> > First, regarding the statement "network interactions based on past ensemble spiking history are hypothesized to be sparse" -- do you have some citations that I could refer to on this point?
> >
> > Second, regarding the statement "optimizing over all model parameters (as would be the case for regularized optimization) becomes intractable as the number of neurons analyzed increases" -- could you provide a quick overview of how the computational complexity grows for the current application? Is it exponential in the number of neurons? Are there other assumptions besides sparsity that could mitigate this?
> >
> > Thank you.

---

> > > ### Author Response · Authors · 2021-08-19
> > > **Additional information about modeling motivations**
> > >
> > > Thank you for your follow-up inquiry about our modeling motivations. First, regarding literature on sparse network interactions, we would like to point out the following representative references
> > > - From statistical modeling and signal processing: GLMs with sparse history coefficients are suitable for explaining the dynamics of neuronal ensembles;
> > >   - I. H. Stevenson et al., "Bayesian inference of functional connectivity and network structure from spikes", *IEEE Trans. Neural Syst. Rehabilitation Eng.* 17(3), pp. 203-213, 2009.
> > >   - D. Pfau et al., "Robust learning of low-dimensional dynamics from large neural ensembles", *Adv. Neural Inform. Process. Syst. 26 (NIPS 2013)*, 2013.
> > >   - L. Aitchison et al., "Model-based Bayesian inference of neural activity and connectivity from all-optical interrogation of a neural circuit", *Adv. Neural Inform. Process. Syst. 30 (NIPS 2017)*, 2017.
> > > - From theoretical neurosciences and biophysical modeling: network models of ensemble activity with sparse coupling among neurons are optimal for information representation;
> > >   - S. Song et al., "Highly nonrandom features of synaptic connectivity in local cortical circuits", *PLoS Biol.* 3(10): e350, 2005.
> > >   - N. Brunel, "Is cortical connectivity optimized for storing information?", *Nat. Neurosci.* 19, pp. 749-755, 2016.
> > >   - A. Litwin-Kumar et al., "Optimal degrees of synaptic connectivity", *Neuron* 93(5), pp. 1153-1164.e7, 2017.
> > > - From electrophysiology: the experimentally measured synaptic connectivity of cortical neurons is sparse;
> > >   - H. Markram et al., "Physiology and anatomy of synaptic connections between thick tufted pyramidal neurones in the developing rat neocortex", *J. Physiol.* 500, pp. 409–440, 1997.
> > >   - A. M. Thomson and A. P. Bannister, "Interlaminar connections in the neocortex", *Cerebral Cortex* 13, pp. 5-14, 2003.
> > >   - C. Holmgren et al., "Pyramidal cell communication within local networks in layer 2/3 of rat neocortex", *J. Physiol.* 551: pp. 139–153, 2003.
> > >
> > > Second, regarding the scaling of the parameters with ensemble size, this relation is at worst exponential. For a set of $C$ neurons, suppose we consider a history integration window with $P$ parameters, and that we observed up to $C^{th}$-order coordinated spiking. Then, the number of parameters for one mark is $C \times P + 1$, and the total number of parameters for the mGLM is $( C \times P + 1) \times (2^{C}-1)$. However, if we observed at most $r^{th}$-order coordinated spiking for a fixed integer $r$, then the total number of parameters for the mGLM is on the order of $( C \times P + 1) \times (2^{r}-1)$. This mitigates the exponential dependence on ensemble size.
> > >
> > > This is an issue shared by maximum entropy models, in the context of which Ganmor et al. \[31\] have proposed a thresholding rule so that only the reliably observed higher-order interactions are modeled. Besides our sparsity assumption, we also adopted this thresholding rule (see lines 124-128) to address the question of tractability, by pruning the mark space of higher-order events that occur too infrequently to be reliably modeled and thereby identifying the highest order, $r$, in a data-driven fashion.

---

### Official Review · Reviewer_T3r8 · 2021-07-17

**Rating:** 7
**Confidence:** 3

**Summary:**

This work develops a set of tools for studying higher order interactions in neural activity recordings. Population neural activity is represented as a marked point process, and fit using a time-varying multinomial GLM. Further, the authors present hypothesis testing tools that compare the fit model with a reduced model that models neurons independently. Specifically, the authors present a Deviance measure that can be used in testing whether r-th order interactions are present in data.

**Ethical Concerns:**

no ethical concerns

**Limitations And Societal Impact:**

no negative societal impact

**Main Review:**

* This paper gives a comprehensive set of tools to study higher order interactions. The tools would be very valuable for computational neuroscience audience. Most of my comments are about making some practical/application aspects clearer in the text.

* While there are no apparent inaccuracies in the text, but the main bulk of paper is heavy in mathematical symbols and equations. Improving the writing could greatly improve the readership and appeal.  Defining the notations in a separate subsection in beginning, and stating all the modeling assumptions (time-varying models, sparsity assumptions, smoothening steps) at same place would help.

* Review of previous works that use marked point process tools for neural activity is missing.

*  For modeling time history dependance, the authors switch from $n^*_t$ (marked representation) to $n_t$  (original neural activity) for computational tractability. This should be explained more carefully since it might have significant implications on results / interpretation.

* How sensitive are measures/models to timing jitter? Small timing changes in individual spikes could change the assigned mark. For example, for two neurons and two nearby time bins, an observation of [(0, 0), (1, 1)] could easily become [(1, 0), (0, 1)] by moving the first neuron's spike ahead in time slightly. Since the mapping from actual spike train to marked point process is highly non-linear, this could significantly affect results of the methods.

* While the presented method gives a statistical test to test if there is rth order interaction in a population of C neurons, it does not identify which neurons are interacting. Repeating the test on all subsets of $r$ neurons might not be ideal. Expanding the discussion around this issue would clarify the application for neuroscientific questions.

* How do the empirical results (especially on human data) change with the smoothening parameter $\beta$ ? Showing dependance on/robustness to $\beta$ would help understand the time-varying model better.

* In general, since the approach uses a time-varying model, the methods presented in the paper do not identify higher order effects that occur at very slow time scales. For example, for the data in Figure 2, considering very large time scales (large $W$ or $\beta$ ~ 1) would show significant r=2 order interaction for the whole duration of experiment. Choice of these hyper-paramters is important for practical application and should be mentioned in text.

* Large differences between history-dependent and independent models is revealing, suggesting that large order interactions in simple models can be attributed to simple mechanisms in slightly more complex models. This is explained in main text associated with Figure 2. However, making these analyses clearer would help with a broader readership, who want to apply these tools to data.


**Time Spent Reviewing:**

6 hours

---

> ### Author Response · Authors · 2021-08-10
> **We will integrate the reviewer's suggestions for clarifying our notation, more clearly discuss motivations for our modeling choices, and address the reviewer's questions about practical aspects.**
>
> - **(Regarding a preliminary subsection for notation and assumptions):** Thank you for the suggestion; we will revise the text accordingly to clarify mathematical notations and modeling assumptions.
> - **(Regarding background of marked point processes for neural activity):** We are happy to include a review of prior applications of marked point process models for neural activity.
> - **(Regarding modeling history dependence in terms of original neural activity):** Our revised text will more carefully motivate the change of notation. Since $ n\_{t} $ can be derived from $ n^{\*}\_{t} $ by a one-to-one linear mapping, there is an equivalence between history parameterizations based on $ n\_{t} $ and on $ n^{\*}\_{t} $. The parameterization based on $n_{t}$ is not only more tractable, but more interpretable in the sense that the dependence on past states of ensemble activity is captured in terms of spiking responses rather than indicator vectors.
> - **(Regarding robustness to timing jitter):** We agree with the reviewer that timing jitter is a key consideration in modeling spiking data as Bernoulli-like observations. The proposed methods' robustness to jitter comes from modeling only the reliable interactions. Marks with few events are indeed susceptible to the artifacts described by the reviewer, but these marks occur unreliably and are thus excluded from analysis.
> - **(Regarding tests of $r^{th}$-order interactions):** Thank you for pointing out this lack of clarity -- we tested simultaneously for all sets of $r^{th}$-order interactions, not single subsets of $r$ neurons. We will ensure the revised text reflects this more clearly.
> - **(Regarding hyperparameters $W$ and $\beta$ ):** We are happy to include supplementary simulations to show the expected effects of varying $W$ and $\beta$. As Reviewer 2 has also suggested, these will include quantitative measures of model goodness of fit. The choice of hyperparameters for our main results will be added to the main text to complement the figure legends.
> - **(Regarding clarification of comparisons between history-dependent and independent models):** Thank you for your suggestion. We will clarify these analyses so that they are more accessible to a broader readership.

---

### Decision · Program_Chairs · 2021-09-27

**Decision:**

Accept (Poster)

**Comment:**

This paper presents a model for multi-neuronal spike trains called the mGLM. The observation in time bin $t$ is a binary vector of length $N$, where $N$ is the number of neurons, and it is modeled as a categorical random variable that can take on one of $2^N$ values. The categorical probabilities are modeled via a GLM, with past spiking and external inputs as covariates. The authors then allow for time-varying parameters and propose an OMP-based fitting procedure.

The reviewers were generally favorable of this paper. However, as Reviewer KgmN pointed out, the core idea of a multinomial GLM to capture dependencies in instantaneous counts is one that has been explored previously (Ba et al, 2014), so the main technical advance here is in the time-varying parameters and the OMP fitting procedure. I'd also note that many have recognized the limitations of conditional independence assumptions in standard GLMs for neural data; indeed this is a main motivation for latent variable models like the Poisson LDS (see Macke et al, NeurIPS 2011, e.g.) and the recurrent linear model (Pachitariu, NeurIPS 2011). I'm also skeptical of the exponential complexity in model parameters in the mGLM, and the time-varying aspect seems to exacerbate the problem.

Despite my reservations, I will go along with the reviewers and recommend acceptance.